# Antimalarial drug resistance molecular makers of *Plasmodium falciparum* isolates from Sudan during 2015–2017

Maazza Hussien[1,2☯], Muzamil Mahdi Abdel Hamid[2☯]*, Elamin Abdelkarim Elamin[1], Abdalla O. Hassan[2], Arwa H. Elaagip[2], Abusofyan Hamattallah A. Salama[3], Mohammed H. Abdelraheem[2,4], Abdelrahim O. Mohamed[5☯]

1 Department of Medical Parasitology and Entomology, Faculty of Medical Laboratory Sciences, Al Neelain University, Khartoum, Sudan, 2 Institute of Endemic Diseases, University of Khartoum, Khartoum, Sudan, 3 Faculty of Medical Laboratory Sciences, North Kordofan University, Alobied, Sudan, 4 Ashworth Laboratories, School of Biological Sciences, University of Edinburgh, Edinburgh, United Kingdom, 5 Department of Biochemistry, Faculty of Medicine, University of Khartoum, Khartoum, Sudan

☯ These authors contributed equally to this work.
* mahdi@iend.org, mahdi@uofk.edu

**Data Availability Statement:** This publication uses data generated using Sudanese samples in collaboration with MalariaGEN SpotMalaria Project (https://www.malariagen.net/projects/spotmalaria).

## Abstract

### Background

Current malaria control and elimination strategies rely mainly on efficacious antimalarial drugs. However, drug resistance is a major threat facing malaria control programs. Determination of drug resistance molecular markers is useful in the monitoring and surveillance of malaria drug efficacy. This study aimed to determine the mutations and haplotypes frequencies of different genes linked with antimalarial drug resistance in certain areas in Sudan.

### Methods

A total of 226 dried blood spots (DBS) of microscopically diagnosed *P. falciparum* isolates were collected from Khartoum and three other areas in Sudan during 2015–2017. *Plasmodium falciparum* confirmation and multiplicity of infection was assessed using the Sanger's 101 SNPs-barcode and speciation was confirmed using regions of the parasite mitochondria. Molecular genotyping of drug resistance genes (*Pfcrt*, *Pfmdr1*, *Pfdhfr*, *Pfdhps*, *exonuclease*, *Pfk13*, parasite genetic background (PGB) (*Pfarps10*, *ferredoxin*, *Pfcrt*, *Pfmdr2*)) was also performed. All genotypes were generated by selective regions amplicon sequencing of the parasite genome using the Illumina MiSeq platform at the Wellcome Sanger Institute, UK then genotypes were translated into drug resistance haplotypes and species determination.

### Findings

In total 225 samples were confirmed to be *P. falciparum*. A higher proportion of multiplicity of infection was observed in Gezira (P<0.001) based on the Sanger 101 SNPs -barcode. The overall frequency of mutant haplotype *Pfcrt* 72–76 CVIET was 71.8%. For *Pfmdr*1, N86Y was detected in 53.6%, Y184F was observed in 88.1% and D1246Y was detected in 1.5%

Sequences data including nucleotide sequencing information were deposited in the public repository European Nucleotide Archive (ENA), with accession numbers provided as supplementary data (S2 Suppl).

**Funding:** The authors received no specific funding for this work.

**Competing interests:** The authors have declared that no competing interests exist.

of the samples. The most frequently observed haplotype was YFD 47.4%. For *Pfdhfr* (codons 51, 59,108,164), the ICNI haplotype was the most frequent (80.7%) while for *Pfdhps* (codons 436, 437, 540, 581, 613) the (SGEAA) was most frequent haplotype (41%). The Quadruple mutation (*dhfr* N51I, S108N + *dhps* A437G, K540E) was the highest frequent combined mutation (33.9%). In *Pfkelch13* gene, 18 non-synonymous mutations were detected, 7 of them were detected in other African countries. The most frequent *Pfk13* mutation was E433D detected in four samples. All of the *Pfk13* mutant alleles have not been reported to belong to mutations associated with delayed parasite clearance in Southeast Asia. PGB mutations were detected only in *Pfcrt* N326S\I (46.3%) and *Pfcrt* I356T (8.2%). The *exonuclease* mutation was not detected. There was no significant variation in mutant haplotypes between study areas.

## Conclusions

There was high frequency of mutations in *Pfcrt*, *Pfdhfr* and *Pfdhps* in this study. These mutations are associated with chloroquine and sulfadoxine-pyrimethamine (SP) resistance. Many SNPs in *Pfk13* not linked with delayed parasite clearance were observed. The *exonuclease* E415G mutation which is linked with piperaquine resistance was not reported.

## Introduction

Malaria is considered a major public health problem in Sudan. According to the 2018 annual estimates, 1,617,499 confirmed malaria cases were reported and the reported deaths were 5,003 [1]. *Plasmodium falciparum* is responsible for 91.2% of malaria cases in the country while *P. vivax* makes 8.8% of the cases in Sudan [1]. The *P. falciparum* infections were most prevalent in eastern Sudan (86.8%), Gezira and Khartoum states (97.9% and 75%, respectively). *Plasmodium vivax* infections represent 12.5% of the total malaria infections in Khartoum and 7.5% in eastern Sudan while mixed infections represent 5.7% in eastern Sudan, 1.1% in central Sudan and 12.5% in Khartoum [2]. Effective malaria treatment, insecticide-treated bed nets, indoor residual spraying, and other vector control measures are the main pillars for malaria control [3].

In sub-Saharan countries including Sudan artemisinin-based combination therapy (ACT) is used as a first-line treatment of uncomplicated *P. falciparum* malaria and intravenous artesunate and quinine are used for the treatment of severe malaria [4].

In Sudan, ACT as artesunate and sulfadoxine-pyrimethamine (AS/SP) was introduced in 2004 for malaria treatment replacing chloroquine which became ineffective for the treatment of uncomplicated malaria [5]. Chloroquine had been used since the1960s for uncomplicated malaria, quinine for severe malaria and sulfadoxine-pyrimethamine for malaria in pregnancy and also as the second line of malaria treatment [6].

AS/SP remained in use until 2017 when it was replaced by Artemether-lumefantrine (AL) as the first line of treatment and dihydroartemisinin piperaquine (DHA-PPQ) as a second line for treatment of uncomplicated malaria [2]. Following the introduction of ACT in Sudan in 2004 malaria mortality declined from 1,814 to 612 deaths in 2011 [7,8]. In the subsequent years, mortality increased steadily reaching 1,446 deaths in 2017 [8].

*Plasmodium falciparum* resistance to antimalarial drugs is a major worldwide challenge for malaria control and elimination. The emergence of chloroquine-resistant parasites began in

Southeast Asia in the 1960s following massive deployment of chloroquine. Similar patterns of resistance to alternative antimalarial drugs, such as sulfadoxine–pyrimethamine were reported in the same region [9].

Clinical *in vivo* efficacy trials are used to monitor the effectiveness of the combination therapies. However, this method does not differentiate between the different components of the combination therapies. Therefore, the study of malaria drug resistance molecular markers can be used to monitor each of the components of the ACT [10].

Chloroquine resistance was reported earlier in Sudan with a high level of the *Pfcrt* K76T mutation [11–13]. There is some evidence in African countries that withdrawal of chloroquine for several years was followed by re-emergence of the wild type 76K [14–16].

Mutations in bifunctional dihydrofolate reductase-thymidylate synthase (*Pfdhfr*) and dihydropteroate synthetase (*Pfdhps*) are linked with SP resistance [17]. In Sudan, mutations in *Pfdhfr* (N51I, S108N) and *Pfdhps* (A436G, K540E) were reported earlier [18,19]. Recently, the emergence of *Pfdhfr* (N59R) and *Pfdhps* (S436A, A581G) with an increased level of the combined mutations and hence the resistance to SP has been reported [20,21].

Mutations in *Pfmdr*1 are linked with many treatment failures in drugs such as mefloquine, chloroquine, amodiaquine, quinine, and halofantrine [22–25]. The mutation in *Pfmdr1* (N86Y) linked with chloroquine failure was reported in different parts of Sudan [11,13,26]. Delayed parasite clearance was first described in Southeast Asia in 2013 and it was associated with mutations in *P. falciparum* kelch propeller protein gene (*Pfk13*) [27].

In Africa, a number of mutations were detected in the *Pfk13* gene but not linked with delayed parasite clearance [28] except one isolate in Equatorial Guinea [27] and another in Uganda [28]. The latter mutations were C580Yand A675V respectively, similar to those associated with delayed parasite clearance in Southeast Asia [29,30]. Two recent studies from Sudan were performed on the *Pfk13*gene, the first one showed a single mutation (A621V) not linked with delayed parasite clearance, while the other study showed no mutations [21]. The parasite genetic background (PGB) is a group of mutations in genes including *Pfarps10*, *ferredoxin*, *Pfcrt and Pfmdr2* which have been associated with samples with *Pfk13* mutations [31].

It has been found that a mutation in *exonuclease* E415G was associated with piperaquine resistance [32]. This study aimed to assess antimalarial drug resistance molecular markers of *P. falciparum* isolates from different areas in Sudan during 2015–2017.

## Materials and methods

### Study design, areas and ethics statement

This study was a descriptive cross sectional study, carried out in four areas in Sudan, Khartoum (KH) central Sudan (15˚37'35.39" N 32˚31'35.39" E), New Halfa (NH) eastern Sudan (15˚32'88" N 35˚59'59.86" E), Gezira state (GS) central Sudan (14˚29'59.99" N 33˚09'60.00" E) and North Kordofan (NK) western Sudan (14˚40'59.99" N 29˚55'59.99" E) during October 2015- November 2017. Malaria transmission in all areas is unstable and seasonal with peaks following the rainy season between September to November. A second peak starts in January and extends up to March. The study areas were selected because they vary in their transmission intensity, epidemiology and entomological characteristics [2].

Samples from Khartoum were collected specifically for this study while the samples from the other sites were archived dried blood spots (DBS) from a previous study at the Institute of Endemic Diseases, University of Khartoum.

Ethical clearance for the Khartoum study was obtained from Al Neelain University Institutional Review Board (IRB No: NU-IRP-16-09-04-1) and for the other areas ethical clearance was obtained from the Institute of Endemic Diseases, Research Ethics Committee, University

of Khartoum (Ref No 2/2014). Informed consent was obtained from each study participant or guardians of children prior to enrolment.

## Study subjects and sample collection

A total of 226 DBS samples (KH n = 158, NH n = 30, GS n = 23, NK n = 15) were collected from subjects attending outpatient clinics with fever and other symptoms suggestive of malaria. Patients diagnosed positive for *P. falciparum* malaria were asked to participate in the study. Those who consented to participate were asked to give blood spots. Blood samples were collected from Fingerpicks (50 µl) on Whatman 3 filter paper (Whatman International Ltd Maidstone, England) using aseptic conditions. The spots were allowed to dry, kept in a sealed plastic bag, numbered and stored at room temperature until used. Diagnosis of malaria was performed by microscopic examination of thick and thin blood smears stained with 10% Giemsa solution according to standard protocols [33].

## DNA extraction, amplicon sequencing, and molecular identification of species, drug resistance alleles, and multiplicity of infection

The DBS samples on filter papers were sent to the Wellcome Sanger Institute for processing, molecular genotyping and analysis. DNA extraction was carried out using the QIAGEN DNA Investigator Kit (No. 56504, Qiagen, Crawley, UK). DNA was eluted in 50 µl TE buffer and stored at − 20 ˚C for later use. Amplicon sequencing was performed on all samples using a custom protocol from the Wellcome Sanger Institute (Jacob et al. manuscript in preparation). In brief, targets for genotyping were identified and multiplex PCR primers were designed using the mPrimer software [34]. Primers were constructed to amplify products with a length of 190–250 bp and were combined into 3 pools. A two-step protocol was used to first amplify the selective regions of the parasite genome from extracted genomic DNA followed by a second PCR to incorporate sequencing and multiplexing adapters [35]. *Plasmodium* species was determined by analyzing highly conserved segments of the parasite mitochondrial genome common to all 5 human infecting species (*P. falciparum*, *P. vivax*, *P. ovale*, *P. malariae*, and *P. knowlesi*) [36]. Two separate 200bp segments (non-genic) were sequenced in multiplex and aligned to the reference sequences of *P. falciparum* and *P. vivax* and non-reference alleles determined which species were within each DNA sample. Quality and purity of amplicons were checked using the Agilent DNA 1000 assay kit on a 2100 Bioanalyzer (Agilent Technology). Batched samples (max n = 384) were sequenced in a single MiSeq lane combining all PCR products [37]. Samples reads were de-plexed using the multiplexing adapters and individual CRAM files were aligned to a modified amplicon reference genome (*Pf*3D7_v3). Genotyping was done using bcftools as well as custom scripts to filter and translate genotypes into drug resistance haplotypes [38]. Sequences data including nucleotide sequencing information were deposited in the public repository European Nucleotide Archive (ENA) with accession numbers provided as supplementary data (S2 File).

The multiplicity of infection (MOI) was determined using COIL and Real McCOIL programs [39,40]. Both programs used the SNP barcode of 101 bi-allelic SNPs genotyped by amplicon sequencing. The MOI was expressed as an integer, which is the estimated number of individual parasites within the sample.

Molecular genotyping of antimalarial resistance genes (*Pfcrt*, *Pfmdr1*, *Pfdhfr*, *PfdhPfs*, *exonuclease*, *Pfk13*, parasitegenetic background (*Pfarps10*, *ferredoxin*, *Pfcrt*, *Pfmdr2*) was also performed at specific loci outlined in the results section. Using the reference genome, SNPs were translated to amino-acids and compared with the relevant literature to determine association with antimalarial drug resistance.

**Table 1. Baseline characteristics of study participants and multiplicity of infection (MOI).**

| | Khartoum (KH) | New Halfa (NH) | Gezira (G) | North Kordofan (NK) | Total |
|---|---|---|---|---|---|
| No of samples | 158 | 30 | 23 | 15 | 226 |
| N (%) male | 101 (63.9%) | 16 (53.3%) | 15 (65.2%) | 9 (60%) | 142 (62%) |
| Median age (years) | 13.7 | 19 | 6.4 | 17.5 | 14.15 |
| Multiclonal infections (n = 21) | 20 (13.4%) | 4 (16%) | 6 (27.2%) | 2 (13.3%) | 32 (15.1%) |
| MOI* | 1.13 | 1.16 | 1.27 | 1.13 | 1.15 |

* $P<0.001$

### Data analysis

Frequencies and percentages of mutant alleles and haplotypes were determined using Statistical Package for Social Sciences version 20 software (SPSS Software, Chicago Inc., USA) for data analysis. The variation of mutant haplotypes between study areas was measured using Chi-square test. Statistical significance level was defined as P-value $\leq 0.05$.

## Results

### *Plasmodium falciparum* identification and multiplicity of infection

Characteristics of the study participants are presented in Table 1. Out of 226 samples diagnosed by microscopic examination as *P. falciparum*, 225 were confirmed to be *P. falciparum* by amplicon sequencing. One sample was identified as *P. vivax* and excluded from further analysis. Multiplicity of infection was successfully determined in 211 samples where (32/211) (15.1%) isolates were found to harbor multiclonal *P. falciparum* infection. The overall mean number of parasite clones was 1.15. A higher proportion of multiplicity of infection was seen in Gezira using either COIL or McCOIL methods ($P<0.001$) based on the Sanger 101 SNP-barcodes (Table 1).

### *Plasmodium falciparum chloroquine resistance transporter* gene (*Pfcrt*)

Out of 225 samples, a total of 213 samples were successfully amplified and sequenced for *Pfcrt*. No mutations were detected at codons 72 and 97. However, 71–73% of the samples harbored the mutant alleles M74I, N75E, K76T, A220S, Q271E and R371I (Table 2).

**Table 2. Frequency of mutant alleles in *Pfcrt* and *Pfmdr1*.**

| Drug resistance maker | Mutant alleles | n (%) |
|---|---|---|
| *Pfcrt* (n = 213) | M74I | 153 (71.8%) |
| | N75E | 153 (71.8%) |
| | K76T | 153 (71.8%) |
| | A220S | 154 (72.3%) |
| | Q271E | 155 (73%) |
| | R371I | 155 (73%) |
| *Pfmdr1* (n = 194) | N86Y | 104 (53.6%) |
| | Y184F | 171 (88.1%) |
| | D1246Y | 3 (1.5%) |

*Pfcrt*: *P. falciparum chloroquine resistance transporter* gene; *Pfmdr1*: *P. falciparum* multidrug resistance protein 1 gene.

**Table 3. Frequency of *Pfcrt* and *Pfmdr1* haplotypes.**

| Drug resistance maker | number of mutations | mutation haplotype | KH n = 149 | NH n = 29 | G n = 21 | NK n = 14 | Total n (%) | P. value |
|---|---|---|---|---|---|---|---|---|
| *Pfcrt* (n = 213) | None | Wild Type CVMNK | 48 (32.2%) | 7 (24.1%) | 5 (23.8%) | 0 (0%) | 60 (28.2%) | 0.17 |
| | Triple | CVIET | 101 (67.8%) | 22 (75.9%) | 16 (76.2%) | 14 (100%) | 153 (71.8%) | 0.17 |
| *Pfmdr1* (n = 194) | None | Wild Type NYD | 9 (6.5%) | 0 (0%) | 4 (19.0%) | 0 (0%) | 13 (6.7%) | |
| | Single | **Y**YD | 6 (4.3%) | 1 (3.8%) | 1 (4.7%) | 1 (12.5%) | 9 (3.1%) | |
| | | N**F**D | 51 (36.7%) | 14 (53.8%) | 9 (42.9%) | 3 (37.5%) | 77 (39.7%) | 0.42 |
| | Double | **YF**D | 70 (50.4%) | 11 (42.3%) | 7 (33.3%) | 4 (50.0%) | 92 (47.4%) | 0.48 |
| | | **Y**Y**Y** | 1 (0.7%) | 0 (0%) | 0 (0%) | 0 (0%) | 1 (0.5%) | |
| | Triple | **Y**F**Y** | 2 (1.4%) | 0 (0%) | 0 (0%) | 0 (0%) | 2 (1%) | |

*Pfcrt*: *P. falciparum chloroquine resistance transporter* gene; *Pfmdr1*: *P. falciparum* multidrug resistance protein 1 gene; KH: Khartoum; NH: New Halfa; G:Gezira; NK: North Kordofan.

Regarding *Pfcrt* haplotypes, 153/213 (71.8%) samples carried the mutant haplotype CV**IET** while the wild haplotype CVMNK was detected in 60/213 (28.2%) of the samples. All samples that harbor the mutant haplotype of *Pfcrt* at positions 72–76 (CV**IET**) were also *Pfcrt* mutant at positions A220S, Q271E and R371I. The frequency of *Pfcrt* mutant haplotype CV**IET** in North Kordofan was 100%, Gezira 76%, New Halfa 75% and Khartoum 67%. The differences in frequencies between the different sites were not statistically significant (Table 3).

### *Plasmodium falciparum* multidrug resistance protein 1 gene *(Pfmdr1)*

A total of 194 samples were successfully amplified and sequenced. The N86Y mutation was detected in 104/194 (53.6%) samples, while Y184F was observed in 171/194 (88.1%) and D1246Y in 3/194 (1.5%) samples. There were no mutations detected in S1034, N1042 and F1226. The frequencies of *Pfmdr*1 haplotypes did not significantly vary between the different studied sites.

*Pfmdr*1 **YF**D haplotype was detected in 92/194 (47.4%) samples while N**F**D haplotype was observed in 77/194 (39.7%) samples (Table 3).

### *Plasmodium falciparum* bifunctional dihydrofolate reductase–thymidylate synthase gene (*Pfdhfr*)

Out of 225 samples, a total of 187 samples were successfully amplified and sequenced for *Pfdhfr*. The N51I mutation was detected in 181/187 (96.7%) samples, while C59R was seen in 30/187 (16%) samples and *Pfdhfr* S108N was observed in 183/187 (97.8%) samples (Table 4). There were no mutations detected in A16 and I164.

**Table 4. Frequency of *Pfdhfr* and *Pfdhps* mutant alleles.**

| | Mutant alleles | Frequency |
|---|---|---|
| *Pfdhfr* (n = 187) | A16V | 0 (0%) |
| | N51I | 181 (96.7%) |
| | C59R | 30 (16%) |
| | S108N | 183 (97.8%) |
| *Pfdhps* (n = 183) | S436A | (4.3%) 8 |
| | A437G | 139 (76%) |
| | K540E | 124 (67.8%) |
| | A581G | 48 (26.2%) |

*Pfdhfr*: *P. falciparum* bifunctional dihydrofolate reductase–thymidylate synthase gene; *Pfdhps*: *P. falciparum* dihydropteroate synthase gene.

**Table 5. Frequency of *Pfdhfr* and *Pfdhps* haplotypes.**

| | Number of mutations | Mutation haplotype | KH n = 134 | NH n = 23 | G n = 19 | NK n = 11 | Total n (%) | P. value |
|---|---|---|---|---|---|---|---|---|
| *Pfdhfr* (n = 187) | None | Wild Type NCSI | 3 (2.3%) | 0 (0%) | 0 (0%) | 0 (0%) | 3 (1.6%) | |
| | | ICSI | 1 (0.8%) | 0 (0%) | 0 (0%) | 0 (0%) | 1 (0.53%) | |
| | | NCNI | 2 (1.5%) | 0 (0%) | 0 (0%) | 0 (0%) | 2 (1.1%) | |
| | Double | **IC**NI | 104 (77.6%) | 20 (86.9%) | 17 (89.4%) | 10 (90.9%) | 151 (80.7%) | 0.37 |
| | | N**R**NI | 1 (0.8%) | 0 (0%) | 0 (0%) | 0 (0%) | 1 (0.53%) | |
| | Triple | **IR**NI | 23 (17.2%) | 3 (13.04%) | 2 (10.5%) | 1 (9.1%) | 29 (15.5%) | 0.77 |
| *Pfdhps* (n = 183) | None | Wild Type SAKAA | 33 (24.6%) | 1 (4.3%) | 3 (15.8%) | 2 (18.2%) | 39 (21.3%) | 0.10 |
| | Single | **A**AKAA | 4 (2.9%) | 0 (0%) | 0 (0%) | 1 (9%) | 5 (%) | |
| | | S**G**KAA | 12 (8.9%) | 1 (4.3%) | 0 (0%) | 0 (0%) | 13(%) | |
| | Double | S**GE**AA | 50 (37.3%) | 14 (60.8%) | 6 (31.6%) | 5 (45.5%) | 75 (41%) | 0.24 |
| | | **AG**KAA | 0 (0%) | 1 (4.3%) | 1 (5.2%) | 0 (0%) | 2 (1.09%) | |
| | Triple | S**GEG**A | 31 (23.1%) | 6 (26.1%) | 9 (47.3) | 3 (27.7%) | 48 (26.2%) | 0.19 |
| | | **AGE**AA | 0 (0%) | 1 (4.3%) | 0 (0%) | 0 (0%) | 1 (%) | |
| *Pfdhps/ Pfdhfr* (n = 183) | None | Wild Type NCSI + SAKAA | 3 (2.3%) | 0 (0%) | 0 (0%) | 0 (0%) | 3 (1.6%) | |
| | | NCNI + SAKAA | 2 (1.5%) | 0 (0%) | 0 (0%) | 0 (0%) | 2 (0.7%) | |
| | Double | **IC**NI +SAKAA | 26 (20.0%) | 1 (4.3%) | 2 (10.5%) | 1 (9.1%) | 30 (16.4%) | 0.18 |
| | | N**R**NI + SAKAA | 1 (0.8%) | 0 (0%) | 0 (0%) | 0 (0%) | 1 (0.54%) | |
| | | ICSI+ **A**AKAA | 1 (0.8%) | 0 (0%) | 0 (0%) | 0 (0%) | 1 (0.54%) | |
| | Triple | **IR**NI +SAKAA | 1 (0.8%) | 0 (0%) | 1 (5.2%) | 1 (9.1%) | 2(1.1%) | |
| | | **IC**NI + **A**AKAA | 3 (2.3%) | 0 (0%) | 0 (0%) | 1 (9.1%) | 4 (2.2%) | |
| | | **IC**NI + S**G**KAA | 7 (5.4%) | 1 (4.3%) | 0 (0%) | 0 (0%) | 8(4.4%) | |
| | Quadruple | **IR**NI+ S**G**KAA | 4 (3.1%) | 0 (0%) | 0 (0%) | 0 (0%) | 4 (2.2%) | |
| | | **IC**NI + S**GE**AA | 40 (30.8%) | 11 (47.8%) | 6 (31.6%) | 5 (45.5%) | 62 (33.9%) | 0.42 |
| | | **IC**NI + **AG**KAA | 0 (0%) | 1 (4.3%) | 1 (5.3%) | 0 (0%) | 2 (1.1%) | |
| | | **IR**NI+**A**AKAA | 1 (0.8%) | 0 (0%) | 0 (0%) | 0 (0%) | 1 (0.54%) | |
| | Quintuple | **IR**NI + S**GE**AA | 10 (7.7%) | 2 (8.7%) | 0 (0%) | 0 (0%) | 12 (6.5%) | |
| | | **IC**NI + **AGE**AA | 0 (0%) | 1 (4.3%) | 0 (0%) | 0 (0%) | 1 (0.54%) | |
| | | **IC**NI + S**GEG**A | 22 (16.9%) | 5 (21.7%) | 8 (42.1%) | 3 (27.3%) | 38 (20.7%) | 0.08 |
| | Sextuple | **IR**NI + S**GEG**A | 9 (6.9%) | 1 (4.3%) | 1 (5.3%) | 0 (0%) | 11 (6%) | |

*Pfdhfr*: *P. falciparum* bifunctional dihydrofolate reductase–thymidylate synthase gene; *Pfdhps*: *P. falciparum* dihydropteroate synthase gene.

Regarding genotyping of *Pfdhfr*, the wild haplotype (NCSI) was detected in 3 samples (1.6%) while double mutations in codon 51 and 108 (**IC**NI) was the most frequent haplotype 151/187 (80.7%). The triple mutant haplotype (**IRN**I) was detected in 29/187 (15.5%) samples (Table 5). There was no significant variation in the frequency of mutant haplotypes between the different areas (Table 5).

### *Plasmodium falciparum* dihydropteroate synthase gene (*Pfdhps*)

Out of 225 samples, 183 samples were successfully amplified and sequenced for *Pfdhps*. The mutation S436A was detected in 8/183 (4.3%) samples, while A437G was observed in 139/183 (76%), K540E in 124/183 (67.8%) and A581G was found in 48/183 (26.2%) samples (Table 4).

Regarding *Pfdhps* haplotypes, wild type SAKAA was detected in 39/183 (21.3%) samples. Double mutations in codons 437 and 540 (S**GE**AA) was seen in 75/183 (41%) samples and the triple mutant haplotype A437G, K540E and A581G (S**GEG**A) was observed in 48/183 (26.2%) samples. There was no significant variation in the frequency of mutant haplotypes between the different areas (Table 5).

## Combined mutations of *Pfdhfr* and *Pfdhps*

Quadruple mutations (*dhfr* N51I, S108N + *dhps* A437G, K540E) were identified in 62/183 (33.9%) samples and Quintuple mutations (*dhfr* N51I, S108N + *dhps* A437G, K540E, A581G) were detected in 38/183 (20.7%) samples. There was no significant variation in the frequencies of mutant haplotypes between the different areas (Table 5).

## *Plasmodium falciparum kelch13* and other resistance genes

Out of 225 samples, a total of 176 samples were successfully amplified and sequenced for *Pfk13*. Twenty-two mutations were found in 25/176 (14.2%) samples, 18 non-synonymous mutations and 4 synonymous mutations. *Pfk13*E433D was the most frequent mutation observed in 4 samples in Khartoum and North Kordofan areas (Table 6).

For the parasite genetic background (PGB) mutations. Out of 225 samples, 207 were successfully amplified and sequenced. No mutations were detected in *arps* (V127, D128), *Pfmdr2* T484 and *ferredoxin* D193Y while 96/207 (46.3%) samples harbor mutations in *Pfcrt* N326S\I and 17/207 (8.2%) *Pfcrt* I356T (Table 6). For *Pf exonuclease* E415, a total of 220 samples were successfully amplified and sequenced. No mutation was detected in *exonuclease* E415.

**Table 6. Frequency of mutant alleles in *Pfk13*.**

|  | Mutation | Frequency | Area |
|---|---|---|---|
| *PfK13* n = 176 | K372K | 1 (0.4%) | Khartoum |
|  | K378R | 1 (0.8%) | Khartoum |
|  | D389H | 1 (0.4%) | Khartoum |
|  | E401Q | 1 (0.4%) | New Halfa |
|  | K430E | 1 (0.4%) | Khartoum |
|  | E433D | 4 (2%) | Khartoum (3) and North Kordofan (1) |
|  | F442F | 1 (0.4%) | North Kordofan |
|  | L488L | 1 (0.4%) | Khartoum |
|  | V494I | 1 (0.4%) | Gezira |
|  | A504T | 1 (0.4%) | Khartoum |
|  | E509E | 1 (0.4%) | Khartoum |
|  | Y541H | 1 (0.4%) | Khartoum |
|  | C542Y | 1 (0.4%) | New Halfa |
|  | W565STOP | 1 (0.4%) | North Kordofan |
|  | R575T | 1 (0.4%) | Khartoum |
|  | E606K | 1 (0.4%) | Khartoum |
|  | G625R | 1 (0.4%) | New Halfa |
|  | A626T | 1 (0.4%) | New Halfa |
|  | Y635C | 1 (0.4%) | Khartoum |
|  | Q661P | 1 (0.4%) | Khartoum |
|  | Q661R | 1 (0.4%) | Khartoum |
|  | T677I | 1 (0.4%) | Khartoum |
|  | Wild type | 198 (87.5) |  |
| PGB (parasite genetic background) n = 207 | *Pfcrt* N326S | 96 (46.3%) | All sites |
|  | *Pfcrt* I356T | 17 (8.2%) | All sites |

*fk13*: *P. falciparum* kelch propeller protein gene; PGB: Parasite genetic background genes.

## Discussion

This study was conducted in three regions in Sudan (Western, Eastern and Central) which are characterized by unstable and seasonal malaria transmission. Molecular markers associated with anti-malarial drug resistance were analyzed.

Mutations of chloroquine resistance transporter gene reported in this study were associated with changes in CVMNK (wild haplotype) to CV**IET** (mutant haplotype) (Table 1). The percentage of CV**IET** in this study ranged between 67.8% in Khartoum to 100% in North Kordofan, in accordance with published studies from Sudan [12,21]. Previous studies reported that CV**IET** is the most widespread resistant haplotype in Africa and Asia, while SVMN**T** is dominant in South America and Oceania [41]. These results show no sign of reduction of chloroquine resistance due to withdrawal of chloroquine as reported earlier that withdrawal of chloroquine would lead to regaining of the sensitivity of malaria parasite to chloroquine [14,42]. This finding may indicate that chloroquine is still being used in these areas.

Mutations in the multidrug resistance gene (*Pfmdr1*) in this study showed that Y184F was the highest frequency (88.1%), with N86Y being intermediate (53.6%) and the lowest D1246Y (1.5%). The mutation at position 86 is associated with chloroquine resistance [25]. The results of this study are in agreement with previously published data in Sudan [11,13,26]. On the other hand, the *Pfmdr1* 86Y mutation and/or the Pfmdr1 86Y-Y184 haplotypes were reported to be significantly associated with increased *in vitro* susceptibility of *P. falciparum* parasites to lumefantrine [43,44].

There were no additional mutations in the multidrug resistance gene detected at positions 1034, 1042 and 1226 in this study, which have been shown to affect the parasite response to mefloquine, chloroquine, quinine, and halofantrine *in vitro* [24,43].

Pyrimethamine drug resistance is strongly-linked with mutations in the bifunctional dihydrofolate reductase-thymidylate synthase [45]. Characterization of resistant parasites is based on the number of mutations they carry (single, double, triple, quadruple mutants), which is taken to be an indicator of the level of resistance to the drug. In this study, the percentage of double mutant haplotype **I**CN**I** was 80.7% and 15.5% for the triple mutant haplotype **IRN**I. The results of these gene mutations were similar to recent studies conducted in Sudan [20,21]. The low prevalence of the triple mutant haplotype **IRN**I may be due to low level of C59R mutation which was reported to be absent in Sudan from isolates in 2003 [18,19]. However, it was first detected in one isolate from a study of 63 Isolates in 2012 [46], since then, the frequency of this mutation has increased gradually.

Results of mutations in dihydropteroate synthase, which mediates sulfadoxine drug resistance [17], showed that the occurrence of double mutant haplotype S**GE**AA was 41% and 26.2% for the triple mutant haplotype S**GEG**A. This result is similar to a previous study conducted in Sudan in 2016 [20]. In contrast, there is a recent study reporting that the most prevalent haplotype is S**G**KAA [21].

Since sulfadoxine and pyrimethamine are mostly used in combination (SP), the combined mutations in dhfr+dhps are important to be documented. The results of the frequency of combined mutation showed that the total number of quadruple, quintuple, and hextuple mutations was 131 (71.5%) which indicates a high level of SP resistance [47]. As SP was used in combination therapy as the first line of treatment in Sudan until March 2017 [2], the continued pressure of SP could explain the increasing resistance of the parasite to the drug.

Several polymorphisms in the *Pfk13* propeller gene have resulted in delayed parasite clearance. The results of *Pfk13* showed the occurrence of 22 mutations in 25 (14.2%) samples, 18 non-synonymous mutations and 4 synonymous mutations. The polymorphisms found in these samples were not similar to reported mutations associated with delayed parasite

clearance in South East Asia. The non-synonymous mutations described in this study are in agreement with several studies conducted in Africa [48–51]. Notably, only two studies in Africa, from Equatorial Guinea [29] and Uganda [30] have reported C580Y and A675V mutations that are similar to those in South East Asia and lead artemisinin resistance [52]. Seven of the 18 non-synonymous mutations reported in this study, were also previously reported in other African countries, namely, V494I in Mozambique, C542Y in Burkina Faso and Zambia, W565Stop Central Africa, R575T Chinese immigrant in Ghana, G625R, A626T in Gabon and Y635C in Rwanda [28,53–56]. The remaining eleven mutations are novel mutations and further in vitro studies are needed to explore their relationships with artemisinin resistance [57].

Mutations of artemisinin resistance genetic background have been analyzed for four genes. Mutations were observed only in one gene *Pfcrt* (N326S, I356T). Mutations in the genetic background genes may have lead collectively to the emergence of K13 gene mutations in South East Asia [31]. As it appears there are mutations in only one member of the group and hence favor of the absence of *Pfk13* mutations.

Next generation sequencing was performed to detect the *exonuclease* gene mutation which has been reported together with plasmepsin II-III amplification to be linked with piperaquine resistance in South East Asia [32]. There was no mutation detected in the *exonuclease* gene in this study. However, we have not performed plasmepsin II-III amplification. Piperaquine is the partner drug of the DHA-PPQ, the second line of treatment in Sudan. Absence of *exonuclease* mutation is encouraging as DHA-PPQ will continue to be effective. Further studies are required to monitor *P. falciparum* plasmepsin genes in Africa.

## Conclusions

This study has shown high frequency of mutations in *Pfcrt*, *Pfdhfr* and *Pfdhps* genes. Quadruple and Quintuple combination of mutations of DHFR and DHPS were highly frequent. The above mutations are associated with chloroquine and sulfadoxine-pyrimethamine (SP) resistance. Many SNPs in *Pfk13* not linked with delayed parasite clearance were observed. *Exonuclease* E415G mutation which is linked with piperaquine resistance was not reported.

## Supporting information

**S1 File.**
(XLSX)

**S2 File.**
(XLSX)

**S3 File.**
(PDF)

**S1 Data.**
(DOCX)

## Acknowledgments

The authors would like to thank with gratitude the MalariaGEN staff of Wellcome Sanger Institute Centre, UK staff of Wellcome Sanger Institute Centre, UK for their kind contribution and proofreading the final manuscript. Further information about MalariaGen can be accessed on https://www.malariagen.net/projects/spotmalaria [malariagen.net]; the project is coordinated by the MalariaGEN Resource Centre with funding from Wellcome (098051, 090770).

MalariaGEN *P. falciparum* data release v6.3 is available online http://lookseq.sanger.ac.uk/mystudies.

## Author Contributions

**Conceptualization:** Maazza Hussien, Elamin Abdelkarim Elamin, Mohammed H. Abdelraheem, Abdelrahim O. Mohamed.

**Data curation:** Maazza Hussien, Muzamil Mahdi Abdel Hamid, Mohammed H. Abdelraheem.

**Formal analysis:** Maazza Hussien, Muzamil Mahdi Abdel Hamid, Mohammed H. Abdelraheem.

**Funding acquisition:** Abdalla O. Hassan.

**Investigation:** Abdalla O. Hassan.

**Methodology:** Maazza Hussien, Abdalla O. Hassan, Abusofyan Hamattallah A. Salama.

**Supervision:** Muzamil Mahdi Abdel Hamid, Elamin Abdelkarim Elamin.

**Validation:** Abdelrahim O. Mohamed.

**Visualization:** Maazza Hussien.

**Writing – original draft:** Maazza Hussien, Muzamil Mahdi Abdel Hamid, Arwa H. Elaagip, Abdelrahim O. Mohamed.

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
