## [Decision Letter · Decision Letter 0]

11 Feb 2020

PONE-D-19-34681

Antimalarial drug resistance molecular makers of Plasmodium falciparum isolates from Sudan during 2015-2017

PLOS ONE

Dear Dr Hamid,

Thank you for submitting your manuscript to PLoS ONE. After careful consideration, we felt that your manuscript requires revision, following which it can possibly be reconsidered. Although your manuscript was of interest to the reviewers, major issues were related to study design and data interpretation.   According to the reviewers, a number of methodological concerns should be clarified including study design, sampling and ethical issues (written informed consent?)  For your guidance, a copy of the reviewers' comments was included below.  

We would appreciate receiving your revised manuscript by March 10. To enhance the reproducibility of your results, we recommend that if applicable you deposit your laboratory protocols in protocols.io, where a protocol can be assigned its own identifier (DOI) such that it can be cited independently in the future. For instructions see: http://journals.plos.org/plosone/s/submission-guidelines#loc-laboratory-protocols

We look forward to receiving your revised manuscript.

Kind regards,

Luzia Helena Carvalho, Ph.D.

Academic Editor

PLOS ONE

Journal Requirements:

2. We note that you are reporting an analysis of a microarray, next-generation sequencing, or deep sequencing data set. PLOS requires that authors comply with field-specific standards for preparation, recording, and deposition of data in repositories appropriate to their field. Please upload these data to a stable, public repository (such as ArrayExpress, Gene Expression Omnibus (GEO), DNA Data Bank of Japan (DDBJ), NCBI GenBank, NCBI Sequence Read Archive, or EMBL Nucleotide Sequence Database (ENA)). In your revised cover letter, please provide the relevant accession numbers that may be used to access these data. For a full list of recommended repositories, see http://journals.plos.org/plosone/s/data-availability#loc-omics or http://journals.plos.org/plosone/s/data-availability#loc-sequencing.

Reviewers' comments:

Reviewer's Responses to Questions

**Comments to the Author**

1. Is the manuscript technically sound, and do the data support the conclusions?

Reviewer #1: Yes

Reviewer #2: Yes

Reviewer #3: Yes

2. Has the statistical analysis been performed appropriately and rigorously? 

Reviewer #1: No

Reviewer #2: Yes

Reviewer #3: Yes

3. Have the authors made all data underlying the findings in their manuscript fully available?

Reviewer #1: Yes

Reviewer #2: Yes

Reviewer #3: Yes

4. Is the manuscript presented in an intelligible fashion and written in standard English?

Reviewer #1: No

Reviewer #2: No

Reviewer #3: Yes

5. Review Comments to the Author

Reviewer #1: Reviewer’s report:

Title of manuscript: Antimalarial drug resistance markers of Plasmodium falciparum isolates from Sudan during 2015-2017

Hussien et al.

General comments:

This manuscript covers an important topic with high relevance in public health. Since the current malaria control/elimination strategies rely to a greater extent on the use of antimalarials, it is important to monitor the performance of these drugs using different methods. The manuscript is well written but needs major improvements before it can be accepted for publication.

Specific comments:

1. Abstract:

a. The background is rather fragmented with two issues which are not well connected. Authors should revise and re-write the first statement into two separate sentences focusing on the significance and impacts of drug resistance on malaria control/elimination; and the use of molecular markers in the surveillance of drug resistance

b. Of the drug resistance genes mentioned on lines 61-63, I would suggest to include the type of drug(s) which are normally affected by mutations in each of the genes.

c. How was multiplicity of infection determined and why? Since this is not reported in the results, authors may omit it in the abstract.

d. The presentation of results with numbers and percentage is a bit confusing. Since authors reported frequency/prevalence of mutant alleles, I would suggest to give the percentage and keep the number of samples with such alleles in brackets. I would also recommend to either use frequency or prevalence of alleles and not both.

e. Conclusion: author report that no evidence of resistance in piperaquine, mefloquine and halofantrine were detected. They should explain the markers used to assess resistance to piperaquine. Since no prior information is given about the levels of drug resistance to chloroquine (CQ) and SP, it is difficult to understand why they think that CQ resistance has persisted and SP resistance has increased. No background information or levels of resistance to these drugs were given.

2. Background:

a. In the first paragraph, authors should provide more information on the epidemiological profile of malaria in Sudan, including the recent status as well as changes in malaria transmission (if any). They should also provide a brief account and history malaria treatment guidelines in Sudan. Such information will help the reader to understand the type of drugs which have been used in the country in the past 1-5 decades and the level of drug pressure exerted on parasite populations.

b. The entire section needs to be re-organized with each paragraph clearly addressing the same theme.

3. Methods:

a. Why these study sites were selected for the study? Was this part of a different study or it was an independent study?

b. What research questions were being addressed in this study and what was the study design?

c. What were the inclusion and exclusion criteria for patients to be included in the study?

d. How were study subjects selected and sampled?

e. How was sample collection done?

f. How were the samples processed and stored in the field and lab before DNA extraction?

g. What was the sample size, overall and in each of the four sites?

h. Please provide the exact duration of sample collection (with start and end dates in months)

i. I would suggest to provide a rationale for selecting the genes which were assessed.

j. How was data analysis done?

4. Results:

a. Please provide a baseline table showing the characteristics and the numbers of patients sampled in each of the four sites

b. What was the mean number of parasite clones (MoI/CoI) per site and overall?

c. Since 149 samples were from one site (Khartoum), I would suggest to group together all samples from all the remaining sites (table 2). Similar stratification should be done for the data presented in table 4. This should allow a meaningful comparison between Khartoum and other sites.

d. I would suggest to give a key under each table to describe all the abbreviations and symbols used.

e. The subheadings on exonuclease E415, k-13 and PBG can be combined into Pfk13 and other genes

5. Discussion and conclusion:

a. Authors mentioned that the results show no signs of reduction in CQ resistance in Sudan. It will be important to give a reason and explain why CQ resistance has persisted in Sudan.

b. Why is SP resistance increasing in Sudan?

c. This study covered for sites but only few samples were collected in the other sites except Khartoum. Authors need to give reasons for this and provide an account of the limitations of this study.

d. If possible, authors should minimize repetition of results in the discussion.

6. Others comments.

a. In the key words, “molecular maker” should be changed to “molecular markers”

b. Authors should revise language and take care of typos

Reviewer #2: Abstract

1) State the method used in establishing multiplicity of infection

2) Sentence, .....from four areas in Sudan from 2015-2017. This sentence is incomplete and also indicate the four areas and how they were selected

3) Highlight the most prevalent K13 mutations

4) Provide results for background mutations

5) Conclusion needs to be revised to reflect the main outcomes of the study

6) Summary of the results on mefloquine, piperaquine and halofantrine resistance are not listed in the abstract and thus should be omitted in the conclusion

7) What were the resistance markers of halofantrine, mefloquine and piperaquine resistance

Introduction

8) line 109, sentence; ...the subsequence years,..... This should be adjusted to ......the subsequent years,.....

9) line 111, sentence;....providing bases for ......., This sentence should be revised to .....providing a basis for....

10) Check the grammar of sentence, line; 118, 119, 122; in addition, this sentences are not clear

11) ACTs introduced in 2004 but treatment protocol change occurring in 2017, seems contradictory, please clarify

12) Cross check on the usage of the term 'artemisinin resistance', this should be adjusted to 'delayed parasite clearance' instead of artemisinin resistance

13) Line 135, 136 provide the reference

14) Line 140-141; is not in line with the title of the manuscript

Materials and Methods

15) How was Plasmodium falciparum confirmed in the study

16) How were the DBS cards prepared, dried, stored. How long was the storage period and how was quality ensured. A study by Schwartrz et al., 2015 showed that storage affect the quality of DNA from DBS

17) Describe the study setting, health facilities where the samples were collected from

18) How was blood collected (quantity)

19) Line 158, define the study participants (sex, age etc)

20) line 159, describe how the Giemsa stain was diluted and how the test was done

21) Line 161-162, provide ethical clearance number and attach the ethical clearance form

22) Provide details of how parasite DNA was extracted and how quality was ensured during the extraction process

23) Provide listing of all the primers and the adapter sequences used in this study. This was not provided and so the accuracy of the genotyping could not be cross checked which is a major limitation of this work

24) Describe the source of the primers used (were the primers from previous studies or you generated them)

25) Data management, Data analysis, are missing and should be included

26) If the primers were generated please describe how this was performed

27) How was the sample size of 226 determined, please justify this sample size. Does this sample size give the study the necessary power to measure the intended outcome?

28) How quality control was ensured in the laboratory tests is not described making it difficult to believe in the findings. Please describe how quality was ensured through out the laboratory processes

Results

29) For all the proportions/prevalence values, please provide the fractions (numerator and denominator)

30) Line 201, which mutations are being referred to ....

31) The number of successful amplifications for K13 mutations is too low, please explain the high observed failure rate

32) What was the most prevalent K13 mutation

Discussion

33) line 258....chloroquine as reported earlier, what does this mean?

34) Please clarify on what you mean on line 262-263

35) There was no intentional screening for markers of resistance to mefloquine, halofantrine and piperaquine and thus this should be removed from the discussion

36) Line 277-284, please explain the result that was found

37) Line 295-296, has any study been done on this before??

38) The result on background mutations need to be explained

Conclusion

39) This should be restricted to genotyping resutls not phenotypic as phenotypic resistance was not measrued

40) What do the authors mean in the sentence, .....good point that this mutation has not been recorded in Sudan....?, Please clarify

Acknowledgement

41) The fact that the DBS were all provided by the department was not mentioned in the methods section, which is a major omission. Please provide details as pointed out in my comments in the methods section. On the ethical point of view what was the purpose of collection of this samples by the department, were they for research and if so please provide ethical clearance of the primary study. Also indicate if the primary study participants consented for their samples to be used in further studies

References

42) The references are adequate, provide the missing references as indicated in earlier sections of my review

Reviewer #3: This is highly validated research for cross-sectional drug resistance markers study in Sudan. Even this ms was not described in vitro of relavant samples and clincal drug sensitivity of patients, it is highly informative for malaria research fields, especially prevalence of drug resistance in one of East Afrian countries.

Major comments>

1. Authors should described the relavant drug resistance and used standard resime drug in clinics of field sites in Sudan for P. falciparum and explained relationship of SNP and used drug currently.

2. What about prevalence of malaria in study areas? Pf only or mixed cases with non-Pf malaria species for treatment. If patients treated and exposured with CQ for Pv previously, how to explain about drug sensitivity?

3. What about in vitro drug sensitivity and/or drug sensitivty clincally for field isolates and relationship with SNP about each drug marker? Please describe in manuscript.

Minor comments>

1. Please correct some of italyic font text and parasite name.

2. Were there any low drug sensitivyt samples from field isolates?

3. Were there any places or patients in local clinics or personal TX still use CQ for Pf instead of ACT Tx?

6. PLOS authors have the option to publish the peer review history of their article (what does this mean?). If published, this will include your full peer review and any attached files.

Reviewer #1: No

Reviewer #2: Yes: Moses Ocan (PhD)

Reviewer #3: Yes: Eun-Taek Han

---

## [Author Response · Author response to Decision Letter 0]

27 Mar 2020

Date: 23-03-2020

Dear Editor,

Thank you for reviewing our manuscript and finding it of interest to PLOSONE when revised. We have revised the manuscript extensively following the reviewers’ recommendations and comments and have provided a separate response. The revised manuscript with changes tracking is been submitted with this response. A clean version with final corrections is provided. 

We have also provided supplementary data to include the sequence accession numbers and our sequence data sets. Also the ethical approvals were provided. We hope that the manuscript is now ready and acceptable for publication in PLOSONE. 

Thanking you,

Dr Muzamil Abdel Hamid and Professor Abdelrahim Mohamed 

University of Khartoum.

---

## [Decision Letter · Decision Letter 1]

14 Apr 2020

PONE-D-19-34681R1

Antimalarial drug resistance molecular makers of Plasmodium falciparum isolates from Sudan during 2015-2017

PLOS ONE

Dear Dr Hamid,

Thank you for submitting your manuscript for review to PLoS ONE. After careful consideration, we feel that your manuscript will likely be suitable for publication if it is revised to address specific points raised now by the reviewers. Specifically, the authors should clarify a couple of topics related to methods and results. A significant number of grammatical errors remained in the revised version of MS. Thus, the language needs to be properly adjusted otherwise it might compromise the publication. At this time, we strongly recommend that the manuscript should go through an in-depth proofreading.

We would appreciate receiving your revised manuscript by May 14. To enhance the reproducibility of your results, we recommend that if applicable you deposit your laboratory protocols in protocols.io, where a protocol can be assigned its own identifier (DOI) such that it can be cited independently in the future. For instructions see: http://journals.plos.org/plosone/s/submission-guidelines#loc-laboratory-protocols

We look forward to receiving your revised manuscript.

Kind regards,

Luzia Helena Carvalho, Ph.D.

Academic Editor

PLOS ONE

Reviewers' comments:

Reviewer's Responses to Questions

**Comments to the Author**

1. If the authors have adequately addressed your comments raised in a previous round of review and you feel that this manuscript is now acceptable for publication, you may indicate that here to bypass the “Comments to the Author” section, enter your conflict of interest statement in the “Confidential to Editor” section, and submit your "Accept" recommendation.

Reviewer #1: All comments have been addressed

Reviewer #2: (No Response)

Reviewer #3: All comments have been addressed

2. Is the manuscript technically sound, and do the data support the conclusions?

Reviewer #1: Yes

Reviewer #2: Yes

Reviewer #3: Yes

3. Has the statistical analysis been performed appropriately and rigorously? 

Reviewer #1: Yes

Reviewer #2: No

Reviewer #3: Yes

4. Have the authors made all data underlying the findings in their manuscript fully available?

Reviewer #1: Yes

Reviewer #2: No

Reviewer #3: Yes

5. Is the manuscript presented in an intelligible fashion and written in standard English?

Reviewer #1: No

Reviewer #2: Yes

Reviewer #3: Yes

6. Review Comments to the Author

Reviewer #1: 1. In the background, I would recommend to replace reference #1 with the most recent WHO report of 2019

2. In the results, lines 225-230, 244-247, and 253-256 present results of hyplotypes in Pfcrt, Pfdhfr and Pfhps, resppectively. Authors should change the presentation from genotypes to haplotypes to be specific.

3. Tables 4 and 5 were interchanged. This need to be corrected

4. The manuscript still needs language corrections which authors might seek support from native English speakers.

Reviewer #2: COMMENTS ON THE REVIEW OF PONE-D-19-3468_R1

Thanks editor for the opportunity given to me to review the above manuscript. From the initial review it is clear that the authors did make significant adjustments however, there are still areas that need to be addressed to make the manuscript suitable for publication. Here below are the highlights of the areas that need to be addressed.

1. The sampling procedure used during field data collection should be described. How individual patients were selected for inclusion into the study

2. Describe the quality control done by the Sanger Institute Welcome Trust

3. Describe the procedure of Nested PCR and illumina Miseq used

4. Comprehensive grammatical review is needed for this manuscript

5. Specify the conserved mitochondrial gene used in confirming P. falciparum parasite infection and provide details of how this was performed

6. What concentration of Giemsa stain was used in the laboratory diagnosis of malaria (line 178-179)

7. Line 193-914; the information provided in this manuscript must be stand alone. Therefore, the authors should provide the specifications/details of all the primers including the manufacturer. All the primers used in the PCR/Sequencing/Multiplex PCR and how they were designed for each of the SNPs assessed in this study

8. Provide reference for the Nested amplification protocol used in this study

9. Provide details of how the raw sequences were analyzed to establish the mutations/SNPs in this study

10. Line 228-229; identify/specify the mutations being referred to

Thanks

Moses Ocan (PhD)

Reviewer #3: All the contents requsted were addressed well about drug resistant gene mutation in Sudan recently and followed from comments of reviewers.

7. PLOS authors have the option to publish the peer review history of their article (what does this mean?). If published, this will include your full peer review and any attached files.

Reviewer #1: Yes: Deus S. Ishengoma

Reviewer #2: Yes: Moses Ocan (PhD)

Reviewer #3: No

---

## [Author Response · Author response to Decision Letter 1]

30 Apr 2020

Dear Luzia Helena Carvalho, Ph.D.

Academic Editor

PLOS ONE

Thank you for the kind feedback including the reviewers’ points. We have extensively revised the manuscript including the grammar and language, including the core methodology and results sections (attached two version: clean and one with track changes). Also, our collaborators at MalariaGen, Sanger Institute, UK have extensively revised the methodology procedures including the SNP genotyping and illumina Miseq procedures, so they have made the required corrections as permitted (upcoming methodology paper will be published by Sanger Institute). Also they have kindly offered final proofreading of the manuscript. We also have revised and answered specific points raised by the two reviewers. 

We have described the procedures used in short as permitted by Sanger group in UK. However, we cannot disclose the full procedures and exact primers (PCR/Sequencing/Multiplex PCR), QC and design and sequences (as requested by reviewer 2) all these will be published in upcoming methodology manuscript in preparation (Jacob et al.) (see attached email reply from Sanger regarding this issue). For more information feel free to contact Dr Jacob Christopher Jacob ecj9@sanger.ac.uk ; Dr. Kimberly Johnson kim.johnson@bdi.ox.ac.uk and head of the group Dr Sonia Concalves sg19@sanger.ac.uk from MalariaGen, Sanger Institute in the UK.

To add our samples performed at Sanger was a small part of the whole malaria spot project at Sanger. For the purpose of this paper we are more interested in the results presenting the frequencies and mutations/haplotypes of antimalarial drug resistance genes in Sudan including Kelch 13 gene which are of paramount importance to our malaria control program. 

Thank you.

Sincerely,

Dr Muzamil Mahdi and Prof Abdelrahim Osman

Parasitology Dept. 

University of Khartoum

Sudan

---

## [Decision Letter · Decision Letter 2]

15 May 2020

PONE-D-19-34681R2

Antimalarial drug resistance molecular makers of Plasmodium falciparum isolates from Sudan during 2015-2017

PLOS ONE

Dear Dr. Hamid,

Thank you for submitting your manuscript for review to PLoS ONE. After careful consideration, we feel that your manuscript will likely be suitable for publication if it is revised to address few points raised now by the reviewer. 

We would appreciate receiving your revised manuscript by  May 30. To enhance the reproducibility of your results, we recommend that if applicable you deposit your laboratory protocols in protocols.io, where a protocol can be assigned its own identifier (DOI) such that it can be cited independently in the future. For instructions see: http://journals.plos.org/plosone/s/submission-guidelines#loc-laboratory-protocols

We look forward to receiving your revised manuscript.

Kind regards,

Luzia Helena Carvalho, Ph.D.

Academic Editor

PLOS ONE

Reviewers' comments:

Reviewer's Responses to Questions

**Comments to the Author**

1. If the authors have adequately addressed your comments raised in a previous round of review and you feel that this manuscript is now acceptable for publication, you may indicate that here to bypass the “Comments to the Author” section, enter your conflict of interest statement in the “Confidential to Editor” section, and submit your "Accept" recommendation.

Reviewer #1: All comments have been addressed

Reviewer #2: All comments have been addressed

2. Is the manuscript technically sound, and do the data support the conclusions?

Reviewer #1: Yes

Reviewer #2: Yes

3. Has the statistical analysis been performed appropriately and rigorously? 

Reviewer #1: Yes

Reviewer #2: Yes

4. Have the authors made all data underlying the findings in their manuscript fully available?

Reviewer #1: Yes

Reviewer #2: Yes

5. Is the manuscript presented in an intelligible fashion and written in standard English?

Reviewer #1: Yes

Reviewer #2: Yes

6. Review Comments to the Author

Reviewer #1: 1. Authors' affiliations should be listed sequentially according the order they appear in the list of authors. This will help to reduce confusion caused by jumping numbers in the affiliations of authors.

2. The information presented on liness 104-107 cannot be prevalence but rather the proportion of infections with Pf, Pv or other species of malaria parasites. Sudan cannot have that high prevelance of parasite infections.

3. On line 180, the word "atteding out patient department" is repeated twice.

4. On line 301, the first statement is incomplete or hanging. I would suggest to merge it with the next statement.

5. Authors should correct minor language problems and typos.

Reviewer #2: All comments raised in the previous review have been addressed by the authors. However, the authors could have provided the details of the primer sets used and their design

7. PLOS authors have the option to publish the peer review history of their article (what does this mean?). If published, this will include your full peer review and any attached files.

Reviewer #1: Yes: Deus S. Ishengoma

Reviewer #2: Yes: Moses Ocan (PhD)

---

## [Author Response · Author response to Decision Letter 2]

30 May 2020

Response Reviewer #1: 

Reviewer #1: 1. Authors' affiliations should be listed sequentially according the order they appear in the list of authors. This will help to reduce confusion caused by jumping numbers in the affiliations of authors.

# Authors' affiliations should be listed sequentially according the order they appear in the list of authors.

2. The information presented on liness 104-107 cannot be prevalence but rather the proportion of infections with Pf, Pv or other species of malaria parasites. Sudan cannot have that high prevelance of parasite infections.

# corrected to proportion of infections.

3. On line 180, the word "atteding out patient department" is repeated twice.

# corrected so as avoid word repetition.

4. On line 301, the first statement is incomplete or hanging. I would suggest to merge it with the next statement.

# The sentence was corrected and completed. 

5. Authors should correct minor language problems and typos.

# The minor language problems and typos were found and corrected and the manuscript was extensively revised. 

Response Reviewer #2: 

Reviewer #2: All comments raised in the previous review have been addressed by the authors. However, the authors could have provided the details of the primer sets used and their design

# Thank you. As stated previously and confirmed by Sanger Institute, all details of primer sets and design will appear in the upcoming methodology paper by the MalariaGen Sanger group. This has been stated in the methods section.

---

## [Decision Letter · Decision Letter 3]

16 Jun 2020

Antimalarial drug resistance molecular makers of Plasmodium falciparum isolates from Sudan during 2015-2017

PONE-D-19-34681R3

Dear Dr. Hamid,

We’re pleased to inform you that your manuscript has been judged scientifically suitable for publication and will be formally accepted for publication once it meets all outstanding technical requirements.

Kind regards,

Luzia Helena Carvalho, Ph.D.

Academic Editor

PLOS ONE

Additional Editor Comments (optional):

Reviewers' comments:

Reviewer's Responses to Questions

**Comments to the Author**

1. If the authors have adequately addressed your comments raised in a previous round of review and you feel that this manuscript is now acceptable for publication, you may indicate that here to bypass the “Comments to the Author” section, enter your conflict of interest statement in the “Confidential to Editor” section, and submit your "Accept" recommendation.

Reviewer #1: All comments have been addressed

Reviewer #2: All comments have been addressed

2. Is the manuscript technically sound, and do the data support the conclusions?

Reviewer #1: Yes

Reviewer #2: Yes

3. Has the statistical analysis been performed appropriately and rigorously? 

Reviewer #1: Yes

Reviewer #2: Yes

4. Have the authors made all data underlying the findings in their manuscript fully available?

Reviewer #1: Yes

Reviewer #2: Yes

5. Is the manuscript presented in an intelligible fashion and written in standard English?

Reviewer #1: Yes

Reviewer #2: Yes

6. Review Comments to the Author

Reviewer #1: All comments have been addressed by the authors and suggestions incorporated in the revision manuscript.

Reviewer #2: The authors have provided adequate responses to the areas of concern raised in the previous version of the manuscript. The manuscript is now ready for publication

7. PLOS authors have the option to publish the peer review history of their article (what does this mean?). If published, this will include your full peer review and any attached files.

Reviewer #1: Yes: Deus S. Ishengoma

Reviewer #2: Yes: Moses Ocan (PhD)

---

## [Editor Report · Acceptance letter]

21 Jul 2020

PONE-D-19-34681R3 

Antimalarial drug resistance molecular makers of Plasmodium falciparum isolates from Sudan during 2015-2017 

Dear Dr. Abdel Hamid:

I'm pleased to inform you that your manuscript has been deemed suitable for publication in PLOS ONE. Congratulations! Your manuscript is now with our production department. 

Kind regards, 

on behalf of

Dr. Luzia Helena Carvalho 

Academic Editor

PLOS ONE